# Accomplishing Sustainability in Manufacturing System for Small and Medium-Sized Enterprises (SMEs) through Lean Implementation

Karishma M. Qureshi [1,*], Bhavesh. G. Mewada [1], Saleh Y. Alghamdi [2], Naif Almakayeel [2], Mohamed Rafik N. Qureshi [2] and Mohamed Mansour [2,3]

1 Department of Mechanical Engineering, Parul Institute of Technology, Parul University, Waghodia 391760, India
2 Department of Industrial Engineering, College of Engineering, King Khalid University, Abha 61421, Saudi Arabia
3 Industrial Engineering Department, College of Engineering, Zagazig University, Zagazig 44519, Egypt
* Correspondence: kariq18@gmail.com

**Abstract:** Business enterprises such as small and medium-sized enterprises (SMEs) play a significant role in economic development but struggle for sustainability. A business enterprise such as a manufacturing unit tries many technological innovations and strategic initiatives to accomplish sustainability in the manufacturing system. Lean manufacturing implementation is one such initiative that helps SMEs manufacture value-added products with increased profitability and waste minimization. However, lean implementation in SMEs is challenging. Hence, it is essential to follow a systematic framework and control the critical success factors (CSFs) in attempting lean implementation. The purpose of this research is to find, evaluate, and rank the CSFs of lean implementation of SMEs so that they may be controlled to accomplish successful lean implementation. The CSFs of lean implementation found by an in-depth assessment of the literature are modeled using the interpretative structural modeling (ISM) approach. MICMAC analysis is also used in classifying and understanding the significance of each lean implementation CSF. ISM and MICMAC provide the relationship modeling to reveal the inter-relationships of each lean implementation CSF. Subsequently, the ISM model is validated using the Delphi technique. The interpretative ranking process (IRP) has been applied to rank the CSFs of lean implementations. The results show that sustainability in a manufacturing system, financial capability, and employee involvement hold significant importance in lean implementations in manufacturing SMEs. Practicing managers may benefit from revisiting their lean implementation plans and respective aligned strategies. They will also be in a position to identify and focus on the scarce resources required for the subsequent lean implementations.

**Keywords:** Delphi technique; interpretative ranking process; interpretive structure modeling (ISM); lean implementation; MICMAC; MSME; small and medium enterprises (SMEs)



## 1. Introduction

Organizations can be broadly classified into small and medium-sized enterprises (SMEs) and large enterprises. SMEs and large enterprises differ in many ways when compared under the criteria of organization structure, organization culture, human resources, and their skills, resources, assets, age, etc. SMEs are classified differently in different territories. Many factors influence this, including the number of people, the scale of the industry, the turnover of the industry, assets, resources, and so on. SMEs are considered the most important economic units [1], helping in economic growth, industrial output, and job generation [2]. According to the World Bank, SMEs account for up to 60% of overall employment. It is also worth noting that these SMEs contribute up to 40% of the nation's

gross domestic product (GDP) (The World Bank). Several sectors of SME business enterprises receive regular government financial assistance to run and expand their operations. Apart from limited access to finance, SMEs also face slow growth due to various challenges of insufficient resources such as skilled human resources, state-of-the-art equipment, revolutionary technology, competitive marketing strategies, research and development efforts, and lack of information technology infrastructure, etc., [3]. SMEs in India are categorized into three categories, namely micro, small, and medium enterprises (MSME) based on their investment, turnover, and number of employees employed. Its contribution is beneficial to socioeconomic growth and vital to the Indian GDP [3]. SMEs of India contribute 6.11% of manufacturing to the GDP and 24.63% of the GDP through service activities, and overall, they contribute 33.4% of India's overall manufacturing output. Further, SMEs contribute 45% of exports and maintains a consistent growth rate of 10% [4].

Similar to SMEs in various countries, Indian SMEs also face stiff challenges. Government pressure and public awareness also force them to have sustainability in their manufacturing process [5]. They face challenges in various areas, such as business sustainability, productivity, and cost-related issues [6]. Lean manufacturing may provide solutions to such challenges. By removing non-value-added activities, lean manufacturing has proven to be beneficial in improving the sustainability, the operational economic performance of manufacturing businesses [7], and world-class performance [8]. Lean strategies are beneficial to both SMEs and large enterprises. However, practice shows that lean production methods and instruments have different applicability to them [9]. Apart from boosting productivity by lowering costs, it also helps in sustainability by improving three pillars (economic, environmental, and social) that help sustainability [10,11]. SMEs have simple systems and procedures which allow them to change quickly to the customers' needs [4,12].

To provide extended profitability and long-term sustainability, SMEs are adopting various manufacturing strategies and initiatives [13]. Since lean manufacturing has become one of the contemporary industry's mainstays [14], it is critical to execute lean to reap the greatest benefits. Lean manufacturing implementation may help SMEs manufacture value-added products with reduced costs and increased profit by eliminating unnecessary waste from the manufacturing process. However, implementing lean in SMEs without understating the role of each lean implementation critical success factor (CSF) will lead to unsuccessful attempts. The role of CSFs in lean implementation is vital and hence must be studied. Lean deployments are difficult for large enterprises and SMEs and are aggravated more among India's manufacturing SMEs in process, process technology, and quality [15]. The lean implementation CSFs must be accomplished because they play a significant role in successful lean implementations in SMEs. Several studies have been reported to identify the CSFs for various sectors of SMEs in different parts of the country. There are few studies found on lean CSF modeling for performance improvement [16–19]. Apart from these, there is no study found to provide a relationship model for lean manufacturing. Hence, it is very significant to study relationship modeling to bridge this gap in the literature to accomplish sustainability in the manufacturing system of SMEs. Thus, based on the above premises, the present research poses the following objectives: (a) to identify the various CSFs of lean implementation in the manufacturing sector of SMEs; (b) to provide relationship modeling using interpretive structure modeling (ISM) and Matrice d'Impacts Croisés Multiplication Appliquésà un Classement (MICMAC meaning cross-impact matrix multiplication applied to classification) analysis; and (c) to provide a ranking using the interpretive ranking process (IRP).

The paper is further organized in the following manner. Section 2 provides a literature review to identify the lean implementation CSFs. It also provides a brief introduction to each lean implementation CSF for the manufacturing sector of SMEs. Section 3 discusses the research methodologies of ISM, MICMAC, and IRP. Section 4 provides results obtained through the systematic application of methodologies. Section 5 documents a detailed discussion of the present research. Lastly, Section 6 provides a conclusion.

## 2. Literature Review

Many researchers have attempted lean, lean six-sigma, and lean-green implementation studies in various SME sectors in various countries. Such a mixed approach of lean with six-sigma and green is evident from the literature review. Various studies led to single or numerous lean methods as shown in Table 1.

**Table 1.** Lean-based approaches in the literature.

| Lean-Based Approaches | References |
|:---:|:---:|
| Lean | [3,20–26] |
| Lean six-sigma | [27–29] |
| Lean-green | [15,30–34] |
| Lean-green-agile | [35] |
| Lean-green six-sigma | [36] |

Alaskari, Ahmad and Pinedo-Cuenca, [37] studied the similarity among the CSFs of lean tools and ERP; they found that company size does not have a significant impact on CSFs of lean tools and EPR systems.

Many researchers attempted the modeling of lean implementation CSFs using analytic hierarchy process (AHP) [38] and fuzzy AHP [39] approaches. Achanga et al. [40] designed a fuzzy logic-based advisory system using data collection from 10 manufacturing SMEs for lean implementation in SMEs. The decision-making system used heuristic rules that enable the postulation of scenarios of lean implementation (Do it, Probably do it, Possibly do it, and Do not do it). Van Landeghem [41] studied the feasibility of lean implementations for much-needed sustainability and concluded that the rate of success of implementation in the industry is overwhelmingly disappointing and lacks systematic documentation and support. Rose et al. [42] focused on feasible lean practices which are required to be implemented to be successful in lean implementation. The best practices suggested were based on three categories; least investment, feasible to apply in SME, and recommended by researchers. Rymaszewska [43] used benchmarking approach and studied the various challenges encountered while implementing lean in SMEs. In general, SMEs lack lean awareness and possess limited knowledge as compared to large enterprises [26].

An evaluation of the literature was conducted to identify the CSFs of lean implementation for SMEs. "workforce skill", "in-house expertise" and "organizational culture" were found to be the most critical success factors for lean manufacturing practices [27] carried out an empirical study and concluded that "workforce skill", "in-house expertise" and "organizational culture" were found to be the most common CSFs. Ref. [29], considered the Indian scenario and collected the most important CSFs "management involvement and commitment"; "customer satisfaction"; "leadership; "cultural change"; "employee satisfaction"; "linking to suppliers"; and "employee relation/empowerment". Many researchers (Refs. [3,14,16,27,39] have stated that lean implementation necessitates "good leadership", "management skills", "knowledge", "financial capability", and "learning skills". Various CSF-based studies undertaken ended with the identification of a few to several CSFs that have varying importance. A comprehensive list of CSFs is discussed in [3,44,45]. The present study has identified the most relevant CSFs in consultation with the expert group. The 20 CSFs are described in Table 2, which are highlighted by the researchers in their respective work.

**Table 2.** Lean implementation CSFs with brief descriptions and references.

| Sr. No. | Lean Implementation CSFs | Descriptions | References |
|---|---|---|---|
| 1 | Top management support and commitment | It is top management's proactive support for the effective implementation of various performance-related measures. | [3,8,18,20,26] |
| 2 | Organization strategies and policies | Organizational strategies and policies will drive employees towards the company's mission and vision. | [46] |
| 3 | Change management and organizational culture | Change management helps individuals or teams prepare for and support organizational change. Organizational culture teaches employee behavior in an organization to inculcate values and traits. | [3,20,26,31,33,47] |
| 4 | Organization structure | It provides the system to fulfill any activities within the system to accomplish the targeted organizational goals. | [3] |
| 5 | effective communication | Effective communication helps to share and utilize the information for better decision-making and performance improvement. | [47,48] |
| 6 | Employee attitude | It has a powerful influence on individual performance and positive relationships with subordinates, colleagues, and superiors, having inculcated values in one job performance. | [3] |
| 7 | Supplier involvement management | Involving suppliers in product design and development, processes, testing, etc., enhances product value addition due to the proactive approach of the supplier in outsourcing management. | [31,49] |
| 8 | Customer focus | Increased customer focus will engage the customer in product design and development to ensure needs and expectations. | [8,49,50] |
| 9 | adoption of soft practices and lean tools | It deals with personal behavior with all stakeholders and enhances sociocultural in an organization, whereas lean tools provide hard practices providing scientific methods/techniques and statistical tools. | [51,52] |
| 10 | Value addition | It refers to adding extra features to the given product or enhancing the economic value that lures the customers, thus enhancing markets share, sales, profit, etc. | [3] |
| 11 | Waste minimization | They are the group of processes and standard practices meant to eliminate waste from the processes or system. | [53] |
| 12 | Sustainability in manufacturing system | It is the self-sufficiency of the manufacturing system for profit and competitiveness. | [10,54] |
| 13 | Employee motivation | Employee motivation is the proactive involvement of employees with creativity and enthusiasm. | [46–48] |
| 14 | Government support | Government supports financial loans, tax benefits, and rules and policies to foster SME development. | [31] |
| 15 | Resource capability | It provides the process resources to meet the requirement of each operation. The creation of resource copiability will help allocate aid in the system. | [3] |
| 16 | Skill and expertise | Skill and expertise are the set of deliverables in fulfilling the activities. | [3,31] |
| 17 | Training and education | Employment training ensures expectations and commitment from the employee towards fulfilling the activities. | [8,26,49] |
| 18 | Financial capability | Financial capabilities ensure management takes proactive investment decisions in strategic planning, employee training, and equipment or consultant hiring. | [3,26,31,47] |
| 19 | Employee involvement | The employee's proactive participation in fulfilling the vision and mission of the organization by undertaking efforts for problem solving, idea building, and applying innovative ideas. | [20,31,33,47,49] |
| 20 | Worker empowerment and engagement | Employee empowerment provides support to employees to take the decision or corrective action, whereas employee engagement is the involvement of employees towards productivity improvement, performance enhancement, and reduction in employee turnover. | [22,55] |

## 3. Research Methodology

Because lean implementation has not been well-researched in the Indian context, we adopted mixed-approaches-based methodologies. This article identifies the lean implementation CSFs through literature review and shortlisted them using the Delphi technique. Further, it relates the identified CSFs using ISM relationship modeling techniques, classifies them by MICMAC, and ranks them using IRP. Thus, a combination of research approaches has been employed. The various research methodology has been used in four different phases, which are depicted in Figure 1. Further, each phase is described in detail as follows.

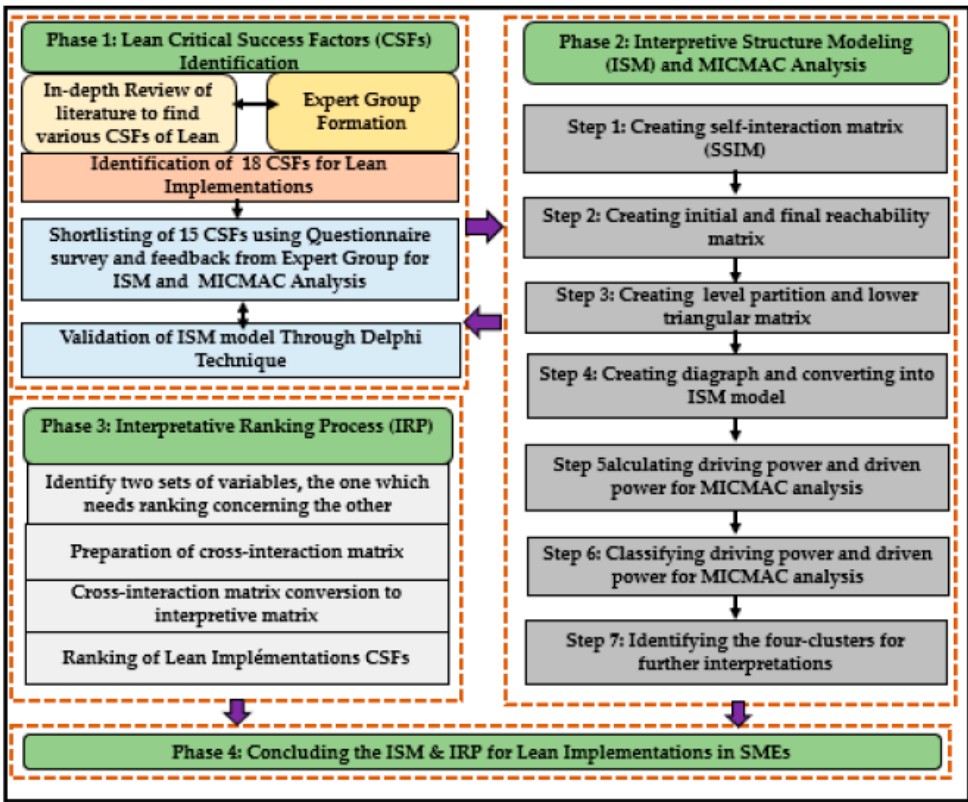

**Figure 1.** The research methodology.

Phase 1 deals with the lean implementation of CSFs identification. The comprehensive review of the literature helps in identifying the lean implementation CSFs in the manufacturing sector. A five-point Likert scale questionnaire consisting of two sections was prepared. The first was based on the general information of the respondent, whereas the second consisted of a selection of lean implementations CSFs. The questionnaire displayed preferences for respondents' feedback, i.e., extremely important, very important, somewhat important, slightly important, and not important, represented by 5, 4, 3, 2, and 1 respectively. The pilot testing of the questionnaire was carried out among a group of academics wherein the questionnaire was discussed to remove any ambiguity and lack of clarity of the questions. The corrected draft was handed over to experts from the SMEs to identify any redundancy or unfamiliarity. To ensure adherence to the university declaration, the Internal Review Board approved the study. Participants agreed to participate in the study and indicated their option to leave at any moment by signing a permission form. Further, they were permitted to refuse to answer any questions. It was agreed upon the confidential use of collected data, with no direct benefit from participation. Participants agreed to audio recording for the interview, anonymous identity, and the ability to retain the original data with the authors. They were further permitted to access collected data at any time, with full freedom to contact any participant. The questionnaire was distributed to engineers, senior engineers, and managers who were connected to the production line. In total, 120 questionnaires were sent through Google Form via email and WhatsApp to the various member SMEs. The members of SMEs were taken from mainly two resources: Gujarat Industrial Development Corporations and the Confederation of Indian Industries. A brief introduction of the research objectives was highlighted at the beginning of the questionnaire. A total of 92 valid responses were received, thus giving an acceptable response rate of 76.66% [56]. Based on the statistical analysis, and discussion with the expert group, 20 CSFs were reduced to 15. The expert group was selected based on their experience, qualifications, and willingness to join the group. Five experts working in the SME manufacturing sector showed their willingness. All of the five experts selected were engineering graduates

and had more than five years of working experience in a lean manufacturing setup. The statistical software package SPSS 26.0 was used to analyze the data.

Phase 2 deals with the ISM and MICMAC modeling. The ISM uses contextual relationship modeling to prepare a self-structural self-interaction (SSIM) matrix, creating reachability and conical matrix and creating digraph and converting into an ISM model. The following steps comprise the ISM and MICMAC methodology. Step 1: Creating SSIM; Step 2: Creating initial and final reachability matrix; Step 3: Create level partition and lower triangular matrix; Step 4: Creating digraph and converting it into an ISM model; Step 5: Calculating driving power and driven power for MICMAC analysis; Step 6: Classifying driving power and driven power for MICMAC analysis; and finally, Step 7: Identifying the four clusters for further interpretations. In creating SSIM, the contractual relationship is considered to explore the interaction between the two lean CSFs let ($x$ and $y$). To represent relationships, 'V' is used when lean CSF $x$ will drive/influence lean CSF $y$; 'A' will be used when lean CSF $y$ will be obtained through lean CSF $x$; 'X' will be used when lean CSF $x$ and lean CSF $y$ help each other, and 'O' will be used when lean CSF $x$ and lean CSF $y$ do not possess any relation. Following the ISM methodological steps, a contextual relationship among lean CSFs yields SSIM. The SSIM has a contextual relationship among lean CSFs identified by the expert group. The initial reachability matrix (IRM). is obtained by transforming SSIM with a binary matrix of 1 and 0. The following rules can be used to replace various symbols such as 'V', 'A', 'X', and 'O'.

–  If the SSIM's ($x, y$) entry is 'V', the reachability matrix's ($x, y$) entry becomes 1 and the ($y, x$) entry becomes 0.
–  If the SSIM's ($x, y$) entry is 'A', the reachability matrix's ($x, y$) entry becomes 0 and the ($y, x$) entry becomes 1.
–  If the SSIM's ($x, y$) entry is 'X', the reachability matrix's ($x, y$) entry becomes 1, and the ($y, x$) entry similarly becomes 1.
–  If the SSIM's ($x, y$) entry is 'O', the reachability matrix's ($x, y$) entry becomes 0, and the ($y, x$) entry similarly becomes 0.

Construct a reachability matrix through SSIM so that the available transitivity is considered. The SSIM matrix may be changed to a reachability matrix using binary numbers (0 and 1) in the reachability matrix. Transitivity may be explored using the rule of CSF if $x > y$ and $y > z$ then x > z wherein ">" provides influence or preference. The reachability element and antecedent element for each CSF can be derived using the final reachability matrix (FRM). It includes the lean CSF itself and another lean CSF that helps. The antecedent elements have their elements as well as other lean CSFs that influence them. The various elements of the iterative process are obtained using the intersection. When the intersection meets such criteria, then the highest level is assigned, and the lean CSF is removed from the iterative process. Such a process will obtain the classification from the highest level to the lowest level. The structural model can be generated using the FRM. Subsequently, a digraph can be realized by eliminating transitivity, as discussed earlier. The lower triangular matrix (LTM) may be used to obtain the digraph that will represent the relationship modeling. The digraph thus obtained provides a directed graph that helps in understanding the role of each CSF.

MICMAC analysis provides a graphical representation of each lean CSFs. It offers an excellent opportunity to study and investigate the relative importance of each lean implementation CSFs. MICMAC analysis helps to classify the lean implementation CSFs into 4 categories. The categories are influenced by the lean implementation CSFs' influence and reliance power. As a result, 4 categories were generated and termed autonomous, dependent, linkage, and independent. The categories generated by MICMAC analysis may also be termed cluster I to cluster IV, respectively.

Phase 3 adopts the IRP to rank the CSFs of lean implementations in SMEs (Sushil, 2009). An interpretative matrix with paired comparison matrix is used by IRP. IRP can nullify the effect of the AHP wherein judgemental bias of an expert may exist, or sometimes it is difficult to make the clear judgment in case of complex hierarchy. Further, the IRP process

warrants the need for interpretive logic for the required dominance of elements between them for each comparison. While carrying out such a comparison, the information for the dominance is not mandatory. IRP also provides the systematic ranking of the CSF based on its outcome. The IRP steps are briefly described in [57–59]. These steps are as follows. (a) Identifying two sets of variables, the one which needs ranking concerning the other. Here, the CSFs of lean implementations for SMEs in the manufacturing sector are being ranked. (b) Preparation of cross-interaction matrix between lean implementation CSFs and lean performance indicators. (c) Cross-interaction matrix conversion to interpretive matrix. (d) Formation of pairwise comparison depending upon interpretive matrix to obtain dominating interactions matrix. (e) Ranking of CSFs and subsequent exploration of dominance and its rank.

Phase 4 deals with the interpretations of ranking derived through ISM and IRP. The conclusion, derived from the ISM and IRP, will be the significant research outcome of the present research.

## 4. Results

The results of each phase are derived and documented as follows.

### 4.1. Phase 1

The mean, standard deviation (SD), and Cronbach's alpha of lean implementation CSFs were calculated and tabulated in Table 3. To maintain the reliability of the questionnaire and simultaneously measure the internal consistency, Cronbach's alpha values were calculated for feedback. The Cronbach's alpha was found within the acceptable limit. Cronbach's alpha > 0.7 provides acceptable internal consistency. The corrected item-total correlation was tested using SPSS 26.0. The five CSFs, namely "organization strategies and policies", "effective communication", "value addition", "resource capability", and "worker empowerment and engagement", were dropped based on the statistical results and consultation with an expert group. Furthermore, the CSFs were renumbered and assigned codes for further analysis.

**Table 3.** Statistical analysis result.

| No. | Critical Success Factors | Code | Mean | SD | Corrected Item-Total Correlation | Cronbach's Alpha |
|---|---|---|---|---|---|---|
| 1 | Organization structure | CSF1 | 4.23 | 0.853 | 0.318 | 0.855 |
| 2 | Financial capability | CSF2 | 4.24 | 0.882 | 0.549 | 0.843 |
| 3 | Government support | CSF3 | 4.15 | 0.960 | 0.314 | 0.856 |
| 4 | Adoption of soft practices and lean tools | CSF4 | 4.16 | 0.929 | 0.296 | 0.857 |
| 5 | Top management support and commitment | CSF5 | 4.02 | 0.937 | 0.471 | 0.847 |
| 6 | Supplier involvement management | CSF6 | 4.09 | 0.885 | 0.514 | 0.845 |
| 7 | Waste minimization | CSF7 | 4.45 | 0.776 | 0.481 | 0.847 |
| 8 | Customer focus | CSF8 | 4.36 | 0.820 | 0.454 | 0.848 |
| 9 | Change management and organizational Culture | CSF9 | 4.13 | 0.880 | 0.684 | 0.836 |
| 10 | Employee attitude | CSF10 | 4.10 | 0.902 | 0.701 | 0.834 |
| 11 | Employee motivation | CSF11 | 4.07 | 0.899 | 0.671 | 0.836 |
| 12 | Skill and expertise | CSF12 | 4.07 | 0.899 | 0.671 | 0.836 |
| 13 | Training and education | CSF13 | 4.04 | 0.888 | 0.414 | 0.850 |
| 14 | Employee involvement | CSF14 | 4.05 | 0.856 | 0.425 | 0.850 |
| 15 | Sustainability in manufacturing system | CSF15 | 4.04 | 0.888 | 0.414 | 0.850 |

### 4.2. Phase 2

The ISM and MICMAC methodologies have been carried out by forming various matrices. Based on the contextual relationship, SSIM was derived by the expert group. Table 4 shows the SSIM derived based on the contextual relationship among various lean CSFs using a set of rules as described in the research methodology. For example, lean CSF1 "organization structure", is compared with lean CSF14 "employee involvement" for their contextual relationship. CSF1 influences CSF14; hence the contextual relationship of 'V' is considered. Similarly, other relationships are completed. Using the binary digits 0 and 1, the 'V', 'A','X', and 'O' were replaced using the rules described in the previous section to obtain the initial reachability matrix (IRM). Table 5 provides the IRM. As per the transitivity

rules, the IRM is transformed into the final reachability matrix (FRM). Table 6 shows the FRM. It also shows the driving power and dependence power obtained by adding the vertical and horizontal total for each CSF.

Table 7 describes the reachability set and antecedent set. Further, the intersection is carried out to reach the level. Based on the data, the CSF15 manufacturing systems sustainability is found to take place at the level i; further, it takes the top position in the ISM. Similarly, repeated iteration will provide various levels. Table 8 shows the iteration results from ii–ix The FRM may lead to obtaining an LTM by rearranging FRM according to the level they are identified, which is shown in Table 7. The structural model is generated using LTM. Based on the level partition, all the CSFs are arranged into the lower triangulation matrix (LTM). Table 9 shows the LTM based on the level partitions. All the CSFs are arranged as per the level partition matrix to obtain the ISM model. All lean implementation CSFs may be plotted as per their driving power and their dependence. An ISM model of lean CSFs is prepared from the digraph, which is presented in Figure 2.

**Table 4.** Self-structural self-interaction (SSIM).

| CSF | Lean Implementation CSFs | 15 | 14 | 13 | 12 | 11 | 10 | 9 | 8 | 7 | 6 | 5 | 4 | 3 | 2 |
|---|---|---|---|---|---|---|---|---|---|---|---|---|---|---|---|
| 1 | Organization structure | V | V | O | O | O | O | O | O | V | O | V | O | V | A | A |
| 2 | Financial capability | V | V | O | O | V | V | V | V | O | V | D | V | A | |
| 3 | Government support | V | V | V | V | V | V | V | V | D | V | V | V | | |
| 4 | Adoption of soft practices and lean tools | V | V | A | V | O | O | A | D | O | V | A | | | |
| 5 | Top management support and commitment | V | V | V | V | V | V | O | V | V | V | | | | |
| 6 | Supplier involvement management | V | O | A | O | A | A | A | V | O | | | | | |
| 7 | Waste minimization | V | O | O | A | A | O | O | O | | | | | | |
| 8 | Customer focus | V | A | O | O | A | A | A | | | | | | | |
| 9 | Change management and organizational culture | V | V | O | O | O | O | | | | | | | | |
| 10 | Employee attitude | V | V | V | O | O | | | | | | | | | |
| 11 | Employee motivation | V | V | V | V | | | | | | | | | | |
| 12 | Skill and expertise | V | O | A | | | | | | | | | | | |
| 13 | Training and education | V | V | | | | | | | | | | | | |
| 14 | Employee involvement | V | | | | | | | | | | | | | |
| 15 | Sustainability in manufacturing system | | | | | | | | | | | | | | |

**Table 5.** Initial reachability matrix.

| CSF | Lean Implementation CSFs | 1 | 2 | 3 | 4 | 5 | 6 | 7 | 8 | 9 | 10 | 11 | 12 | 13 | 14 | 15 |
|---|---|---|---|---|---|---|---|---|---|---|---|---|---|---|---|---|
| 1 | Organization structure | 1 | 0 | 0 | 1 | 0 | 1 | 0 | 1 | 0 | 0 | 0 | 0 | 0 | 1 | 1 |
| 2 | Financial capability | 1 | 1 | 0 | 1 | 1 | 1 | 0 | 1 | 1 | 1 | 1 | 0 | 0 | 1 | 1 |
| 3 | Government support | 1 | 1 | 1 | 1 | 1 | 1 | 1 | 1 | 1 | 1 | 1 | 1 | 1 | 1 | 1 |
| 4 | Adoption of soft practices and lean tools | 0 | 0 | 0 | 1 | 0 | 1 | 0 | 1 | 0 | 0 | 0 | 1 | 0 | 1 | 1 |
| 5 | Top management support and commitment | 0 | 0 | 0 | 1 | 1 | 1 | 1 | 1 | 0 | 1 | 1 | 1 | 1 | 1 | 1 |
| 6 | Supplier involvement management | 0 | 0 | 0 | 0 | 0 | 1 | 0 | 1 | 0 | 0 | 0 | 0 | 0 | 0 | 1 |
| 7 | Waste minimization | 0 | 0 | 0 | 0 | 0 | 0 | 1 | 0 | 0 | 0 | 0 | 0 | 0 | 0 | 1 |
| 8 | Customer focus | 0 | 0 | 0 | 0 | 0 | 0 | 0 | 1 | 0 | 0 | 0 | 0 | 0 | 0 | 1 |
| 9 | Change management and organizational culture | 0 | 0 | 0 | 1 | 0 | 1 | 0 | 1 | 1 | 0 | 0 | 0 | 0 | 1 | 1 |
| 10 | Employee attitude | 0 | 0 | 0 | 0 | 0 | 1 | 0 | 1 | 0 | 1 | 0 | 0 | 1 | 1 | 1 |
| 11 | Employee motivation | 0 | 0 | 0 | 0 | 0 | 1 | 1 | 1 | 0 | 0 | 1 | 1 | 1 | 1 | 1 |
| 12 | Skill and expertise | 0 | 0 | 0 | 0 | 0 | 0 | 1 | 1 | 1 | 1 | 1 | 1 | 0 | 0 | 1 |
| 13 | Training and education | 0 | 0 | 0 | 1 | 0 | 1 | 0 | 0 | 0 | 0 | 0 | 1 | 1 | 1 | 1 |
| 14 | Employee involvement | 0 | 0 | 0 | 0 | 0 | 0 | 0 | 1 | 0 | 0 | 0 | 0 | 0 | 1 | 1 |
| 15 | Sustainability in manufacturing system | 0 | 0 | 0 | 0 | 0 | 0 | 0 | 0 | 0 | 0 | 0 | 0 | 0 | 0 | 1 |

**Table 6.** Final reachability matrix.

| CSF | Lean Implementation CSFs | 1 | 2 | 3 | 4 | 5 | 6 | 7 | 8 | 9 | 10 | 11 | 12 | 13 | 14 | 15 | Total Driving Power |
|---|---|---|---|---|---|---|---|---|---|---|---|---|---|---|---|---|---|
| 1 | Organization structure | 1 | 0 | 0 | 1 | 0 | 1 | 1 * | 1 | 0 | 0 | 0 | 1 * | 0 | 1 | 1 | 8 |
| 2 | Financial capability | 1 | 1 | 0 | 1 | 1 | 1 | 1 * | 1 | 1 | 1 | 1 | 1 * | 1 * | 1 | 1 | 14 |
| 3 | Government support | 1 | 1 | 1 | 1 | 1 | 1 | 1 | 1 | 1 | 1 | 1 | 1 | 1 | 1 | 1 | 15 |
| 4 | Adoption of soft practices and lean tools | 0 | 0 | 0 | 1 | 0 | 1 | 1 * | 1 | 0 | 0 | 0 | 1 | 0 | 1 | 1 | 7 |
| 5 | Top management support and commitment | 0 | 0 | 0 | 1 | 1 | 1 | 1 | 1 | 0 | 1 | 1 | 1 | 1 | 1 | 1 | 11 |
| 6 | Supplier involvement management | 0 | 0 | 0 | 0 | 0 | 1 | 0 | 1 | 0 | 0 | 0 | 0 | 0 | 0 | 1 | 3 |
| 7 | Waste minimization | 0 | 0 | 0 | 0 | 0 | 0 | 1 | 0 | 0 | 0 | 0 | 0 | 0 | 0 | 1 | 2 |
| 8 | Customer focus | 0 | 0 | 0 | 0 | 0 | 0 | 0 | 1 | 0 | 0 | 0 | 0 | 0 | 0 | 1 | 2 |
| 9 | Change management and organizational culture | 0 | 0 | 0 | 1 | 0 | 1 | 1 * | 1 | 1 | 0 | 0 | 1 * | 0 | 1 | 1 | 8 |
| 10 | Employee attitude | 0 | 0 | 0 | 1 * | 0 | 1 | 1 * | 1 | 0 | 1 | 0 | 1 * | 1 | 1 | 1 | 9 |
| 11 | Employee motivation | 0 | 0 | 0 | 1 * | 0 | 1 | 1 | 1 | 0 | 0 | 1 | 1 | 1 | 1 | 1 | 9 |
| 12 | Skill and expertise | 0 | 0 | 0 | 0 | 0 | 0 | 1 | 1 | 1 | 1 | 1 | 1 | 0 | 0 | 1 | 3 |
| 13 | Training and education | 0 | 0 | 0 | 1 | 0 | 1 | 1 * | 1 * | 0 | 0 | 0 | 1 | 1 | 1 | 1 | 8 |
| 14 | Employee involvement | 0 | 0 | 0 | 0 | 0 | 0 | 0 | 1 | 0 | 0 | 0 | 0 | 0 | 1 | 1 | 3 |
| 15 | Sustainability in manufacturing system | 0 | 0 | 0 | 0 | 0 | 0 | 0 | 0 | 0 | 0 | 0 | 0 | 0 | 0 | 1 | 1 |
| | Total (Dependence) | 3 | 2 | 1 | 9 | 3 | 10 | 11 | 12 | 3 | 4 | 4 | 10 | 6 | 10 | 15 | |

*: CSF has transitivity.

**Table 7.** First iteration of lean implementation CSFs level iteration i.

| CSF | Reachability Set | Antecedent Set | Intersection Set | Level |
|---|---|---|---|---|
| CSF1 | 1, 4, 6, 7, 8, 12, 14, 15 | 1, 2, 3 | – | – |
| CSF 2 | 1, 2, 4, 5, 6, 7, 8, 9, 10, 11, 12, 13, 14, 15 | 2, 3 | – | – |
| CSF 3 | 1, 2, 3, 4, 5, 6, 7, 8, 9, 10, 11, 12, 13, 14, 15 | 3 | – | – |
| CSF 4 | 4, 6, 7, 8, 12, 14, 15 | 1, 2, 3, 4, 5, 9, 10, 11, 13 | – | – |
| CSF 5 | 4, 5, 6, 7, 8, 10, 11, 12, 13, 14, 15 | 2, 3, 5 | – | – |
| CSF 6 | 6, 8, 15 | 1, 2, 3, 4, 5, 6, 9, 10, 11, 13 | – | – |
| CSF 7 | 7, 15 | 2, 3, 4, 5, 7, 11, 12, 13 | – | – |
| CSF 8 | 8, 15 | 1, 2, 3, 4, 5, 6, 8, 9, 10, 11, 13, 14 | – | – |
| CSF 9 | 4, 6, 8, 9, 12, 14, 15 | 2, 3, 9 | – | – |
| CSF 10 | 4, 6, 8, 10, 12, 13, 14, 15 | 2, 3, 5, 10 | – | – |
| CSF 11 | 4, 6, 7, 8, 11, 12, 13, 14, 15 | 2, 3, 5, 10, 11 | – | – |
| CSF 12 | 7, 12, 15 | 1, 2, 3, 4, 5, 9, 10, 11, 12, 13 | – | – |
| CSF 13 | 4, 6, 7, 8, 12, 13, 14, 15 | 2, 3, 5, 10, 11, 13 | – | – |
| CSF 14 | 8, 14, 15 | 1, 2, 3, 4, 5, 9, 10, 11, 13, 14 | – | – |
| CSF 15 | 15 | 1, 2, 3, 4, 5, 6, 7, 8, 9, 10, 11, 12, 13, 14, 15 | 15 | I |

Clusters I to IV are obtained in MICMAC analysis as shown in Figure 3. The MICMAC classifies CSF4 "adoption of soft practices and lean tools", CSF6 "supplier involvement management", CSF7 "waste minimization", CSF8 "customer focus", CSF12 "skill and expertise", CSF14 "employee involvement ", and CSF15 "sustainability in manufacturing system" as dependent CSFs. The MICMAC classifies CSF1 "organization structure", CSF2 "financial capability", CSF3 "government support", CSF5 "top management support and commitment", CSF9 "change management and organizational culture", CSF10 "employee attitude", CSF11 "employee motivation", and CSF13 "training and education" are independent CSFs.

**Table 8.** Iteration results (ii–ix).

| Iteration | CSFs | Reachability Set | Antecedent Set | Intersection Set | Level |
|---|---|---|---|---|---|
| ii | 7 | 7 | 2, 3, 4, 5, 7, 11, 12, 13 | 7 | II |
| ii | 8 | 8 | 1, 2, 3, 4, 6, 8, 9, 10, 11, 13, 14 | 8 | II |
| iii | 6 | 6 | 1, 2, 3, 4, 5, 6, 9, 10, 11, 13 | 6 | III |
| iii | 12 | 12 | 1, 2, 3, 4, 5, 9, 10, 11, 12, 13 | 12 | III |
| iii | 14 | 14 | 1, 2, 3, 4, 5, 9, 10, 11, 13, 14 | 14 | III |
| iv | 4 | 4 | 1, 2, 3, 4, 5, 9, 10, 11, 13 | 4 | IV |
| v | 1 | 1 | 1, 2, 3 | 1 | V |
| v | 9 | 9 | 2, 3, 9 | 9 | V |
| v | 13 | 13 | 2, 3, 5, 10, 11, 13 | 13 | V |
| vi | 10 | 10 | 2, 3, 5, 10 | 10 | VI |
| vi | 11 | 11 | 2, 3, 5, 10, 11 | 11 | VI |
| vii | 5 | 5 | 2, 3, 5 | 5 | VII |
| viii | 2 | 2 | 2, 3 | 2 | VII |
| ix | 3 | 3 | 3 | 3 | IX |

**Table 9.** Lower triangulation matrix (LTM).

| CSF | Lean Implementation CSFs | 15 | 7 | 8 | 6 | 12 | 14 | 4 | 1 | 9 | 13 | 10 | 11 | 5 | 2 | 3 |
|---|---|---|---|---|---|---|---|---|---|---|---|---|---|---|---|---|
| 15 | Sustainability in manufacturing system | 1 | 0 | 0 | 0 | 0 | 0 | 0 | 0 | 0 | 0 | 0 | 0 | 0 | 0 | 0 |
| 7 | Waste minimization | 1 | 1 | 0 | 0 | 0 | 0 | 0 | 0 | 0 | 0 | 0 | 0 | 0 | 0 | 0 |
| 8 | Customer focus | 1 | 0 | 1 | 0 | 0 | 0 | 0 | 0 | 0 | 0 | 0 | 0 | 0 | 0 | 0 |
| 6 | Supplier involvement management | 1 | 0 | 1 | 1 | 0 | 0 | 0 | 0 | 0 | 0 | 0 | 0 | 0 | 0 | 0 |
| 12 | Skill and expertise | 1 | 1 | 1 | 0 | 1 | 0 | 0 | 0 | 1 | 0 | 1 | 1 | 0 | 0 | 0 |
| 14 | Employee involvement | 1 | 0 | 1 | 0 | 0 | 1 | 0 | 0 | 0 | 0 | 0 | 0 | 0 | 0 | 0 |
| 4 | Adoption of soft practices and lean tools | 1 | 1 | 1 | 1 | 1 | 1 | 1 | 0 | 0 | 0 | 0 | 0 | 0 | 0 | 0 |
| 1 | Organization structure | 1 | 1 | 1 | 1 | 1 | 1 | 1 | 1 | 0 | 0 | 0 | 0 | 0 | 0 | 0 |
| 9 | Change management and organizational culture | 1 | 1 | 1 | 1 | 1 | 1 | 1 | 0 | 1 | 0 | 0 | 0 | 0 | 0 | 0 |
| 13 | Training and education | 1 | 1 | 1 | 1 | 1 | 1 | 1 | 0 | 0 | 1 | 0 | 0 | 0 | 0 | 0 |
| 10 | Employee attitude | 1 | 1 | 1 | 1 | 1 | 1 | 1 | 1 | 0 | 1 | 1 | 0 | 0 | 0 | 0 |
| 11 | Employee motivation | 1 | 1 | 1 | 1 | 1 | 1 | 1 | 0 | 0 | 1 | 0 | 1 | 0 | 0 | 0 |
| 5 | Top management support and commitment | 1 | 1 | 1 | 1 | 1 | 1 | 1 | 1 | 0 | 1 | 1 | 1 | 1 | 0 | 0 |
| 2 | Financial capability | 1 | 1 | 1 | 1 | 1 | 1 | 1 | 1 | 1 | 1 | 1 | 1 | 1 | 1 | 0 |
| 3 | Government support | 1 | 1 | 1 | 1 | 1 | 1 | 1 | 1 | 1 | 1 | 1 | 1 | 1 | 1 | 1 |

ISM model validation through the Delphi technique: The formulated model was reviewed and validated using the Delphi technique, as shown in Figure 4. Three Delphi members (SME entrepreneurs from the solenoid valve manufacturing unit, boiler manufacturing unit, and casting machining unit) not related to the expert group were consulted. Later on, they agreed to participate anonymously. The feedback after each round is discussed as follows. Feedback from the first round: Out of 20 lean implementation CSFs criteria, Delphi members were asked to drop 5 criteria. Delphi 1 and Delphi 3 dropped "value addition" (mean 2.5), "effective communication" (mean 2.9), "resource capability" (mean 2.8), "worker empowerment and engagement" (mean 2.6), and "organization strategies and policies" (mean 2.7). Feedback from the second round: All three members were sent the contextual relationship matrix for verification. Out of 105 pairwise comparisons, the feedback of Delphi 1, Delphi 2, and Delphi 3 did not match with the expert group in pairwise comparisons. Later on, a consensus was reached. Feedback from the third round: All Delphi members were sent the final ISM model with a MICMAC analysis diagram. All of them agreed on the ISM model, and a consensus was reached.

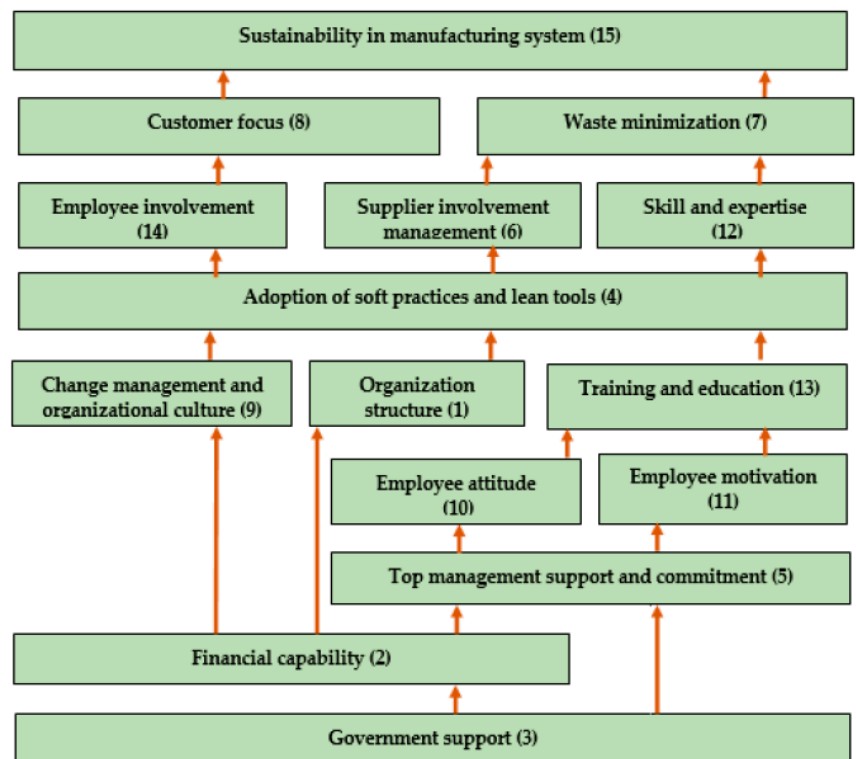

**Figure 2.** ISM for Lean Implementations CSFs.

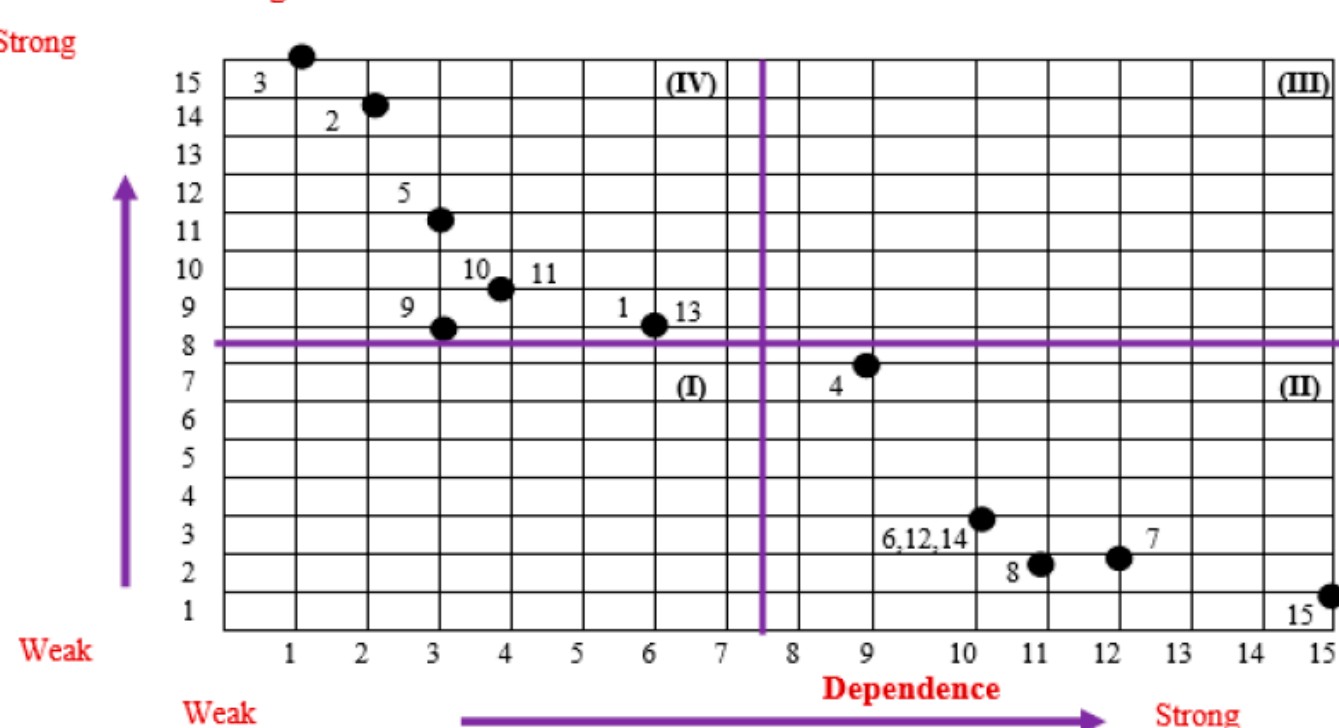

(Legend: Lean implementation CSFs classification I: Autonomous II: Dependent, III: Linkage and IV: Independent.)

**Figure 3.** Driving power and dependence diagram using MICMAC.

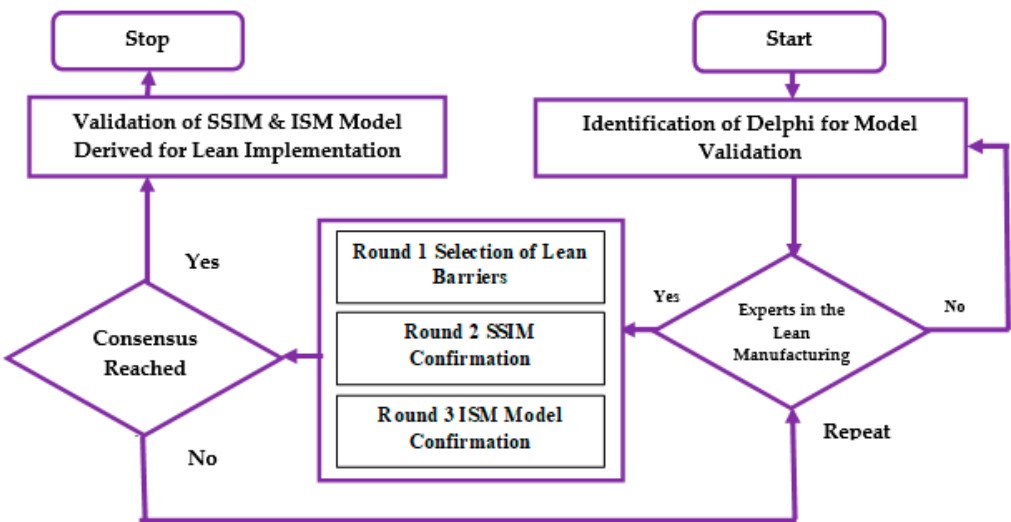

**Figure 4.** Validation of ISM model through Delphi technique.

### 4.3. Phase 3

To accomplish the IRP of lean implementation, eight relevant performance criteria have been identified. P1 to P8 performance criteria were selected by reviewing the literature. The P1 to P8 selected are "quality management" (P1), "production volume linked productivity" (P2), "manufacturing lead-time" (P3), "product-related design and development process" (P4), "profitability" (P5), "brand value in the market" (P6), "market share" (P7), and "customer satisfaction" (P8). A binary value of 1 or 0 was used to represent the existence or non-existence of the relationship between lean implementation CSFs and lean performance criteria. Table 10 shows an interaction matrix. The lean implementation CSFs and lean performance criteria are compared using the contextual relationship from the cross-interpretive matrix. Table 11 depicts the interpretive matrix showing the relationship between the lean implementation CSFs and lean performance criteria. The lean knowledge base matrix as shown in Table 12, helps in developing dominating and nondominating lean implementation CSFs concerning lean performance criteria, P1 to P8. Table 13 shows the dominating interaction matrix. Table 14 shows the dominance matrix. The influence of one variable on the other can be represented diagrammatically. Figure 5 shows an interpretive ranking model.

**Table 10.** Cross-interaction matrix of CSFs and lean performance criteria.

| Lean CSFs | P1 | P2 | P3 | P4 | P5 | P6 | P7 | P8 |
|---|---|---|---|---|---|---|---|---|
| CSF1 | 1 | 1 | 0 | 0 | 0 | 0 | 1 | 1 |
| CSF2 | 0 | 1 | 0 | 0 | 1 | 1 | 0 | 0 |
| CSF3 | 1 | 0 | 0 | 0 | 0 | 1 | 0 | 1 |
| CSF4 | 1 | 1 | 0 | 1 | 0 | 0 | 0 | 1 |
| CSF5 | 0 | 1 | 1 | 0 | 0 | 0 | 0 | 1 |
| CSF6 | 1 | 1 | 1 | 0 | 0 | 0 | 0 | 1 |
| CSF7 | 0 | 1 | 0 | 0 | 1 | 1 | 1 | 0 |
| CSF8 | 0 | 1 | 0 | 0 | 1 | 0 | 0 | 0 |
| CSF9 | 0 | 1 | 0 | 0 | 0 | 1 | 1 | 0 |
| CSF10 | 0 | 1 | 0 | 0 | 0 | 0 | 0 | 1 |
| CSF11 | 1 | 0 | 0 | 0 | 0 | 0 | 0 | 1 |
| CSF12 | 0 | 1 | 0 | 1 | 0 | 0 | 0 | 0 |
| CSF13 | 0 | 1 | 1 | 0 | 0 | 0 | 0 | 0 |
| CSF14 | 0 | 0 | 0 | 1 | 0 | 0 | 1 | 1 |
| CSF15 | 0 | 1 | 0 | 1 | 1 | 0 | 0 | 1 |

**Table 11.** Interpretive matrix.

| CSFs/Performance Criteria | Quality Management | Production Volume Linked Productivity | Manufacturing Lead-Time | Product-Related Design and Development Process | Profitability | Brand Value in the Market | Market Share | Customer Satisfaction |
|---|---|---|---|---|---|---|---|---|
| | P1 | P2 | P3 | P4 | P5 | P6 | P7 | P8 |
| CSF1 | Relationship between higher authority and workers leads to good-quality products | Its interaction and continuous interaction motivate the employee to complete the work fast | | | | | It promotes employee empowerment and improves levels of job satisfaction leading to a higher market position | A healthy and supportive environment leads to good service, on-time delivery leads to customer satisfaction |
| CSF2 | | It increases the efficiency of the production, saving money on materials | | | Focusing more on waste reduction rather than waste disposal can save more money on materials, packaging, and technology | Promoting minimum wastage, protecting the environment, and promoting green increase its brand value | | |
| CSF3 | Governments' standardization and policies help in producing good quality products | | | | | Governments provide the guidelines which help to maintain goods and services and increase brand value and promotion of brands | | Standardized government products earn customer's trust |
| CSF4 | Identifying all types of waste, product defects and eliminating them leads to good quality products with no compromise | It increases the flow of information and products with reduced waiting times | | All level employees work together to improve or make a new product design | | | | Eliminating non-value-added activity leads to customer satisfaction |
| CSF5 | | Determines the flow of information at all levels within the company, thus improving communication, increasing efficiency | Removes duplication of work, more excellent employee performance, faster decisions making, reduces lead time | | | | | Promotes better communication and transparency, better after services, leading to satisfied customers |
| CSF6 | They help in finding better options for the raw materials, thus helping in maintaining a low inventory level | It helps in boosting the production and saves the company costs | It shortens the lead time | | | | | Good quality goods and services lead to satisfied customers |
| CSF7 | | Prevent the employees from making small mistakes, thus prevent from time wastage | | | Educated workers increase the profit because of their excellent skills and education, leading to better performance | Employees constantly progressing gives the company more value | Educated and skilled employees increase the market value | |

**Table 11.** *Cont.*

| CSFs/Performance Criteria | Quality Management | Production Volume Linked Productivity | Manufacturing Lead-Time | Product-Related Design and Development Process | Profitability | Brand Value in the Market | Market Share | Customer Satisfaction |
|---|---|---|---|---|---|---|---|---|
| | P1 | P2 | P3 | P4 | P5 | P6 | P7 | P8 |
| CSF8 | | Employees' performance, commitment, and involvement increase the productivity | | | It is an influential and essential tool that, when managed well, can lead to an increase in profitability | | | |
| CSF9 | | It helps in reducing cost | | | | Lowers the risk in the supply chain and increases brand value | Sustainability helps in increasing the market value | |
| CSF10 | | It leads to creativity and innovation that increase the productivity | | | | | | Good employee, better service, and helpfulness lead to customer satisfaction |
| CSF11 | Good quality of raw materials leads to good quality products | | | | | | | More options and varieties for the customers make customers happy |
| CSF12 | | Motivated and happy employees help increase the productivity | | Employee involvement leads to design improvement/new design | | | | |
| CSF13 | | Educated and trained employees work efficiently and perform their daily jobs well | Skill workers reduce the lead time | | | | | |
| CSF14 | | | | To understand the customer needs and help them honestly | Loyal customers help save costs and increase profits | | Happy customers promote the brand increasing its market value | Enhancing customer satisfaction and building customer relationship |
| CSF15 | | Motivated employees can increase productivity by increasing the production and higher quantity of work | | Motivated employees create new designs and products with their talents and creativity | Employee commitment and company loyalty lead to profitability | | | |

**Table 12.** Lean knowledge base using interpretive logic.

| Dominance Comparison of CSFs | Performance Indicator(s) Influenced | Dominance Comparison of CSFs | Performance Indicator(s) Influenced |
|---|---|---|---|
| 1 Dominating 2 | P2 | 8 Dominating 11 | P5 |
| 1 Dominating 3 | P1 | 8 Dominating 12 | P2 |
| 1 Dominating 6 | P1, P2, P8 | 9 Dominating 1 | P2, P7 |
| 1 Dominating 11 | P1, P8 | 9 Dominating 2 | P2, P6 |
| 1 Dominating 14 | P7, P8 | 9 Dominating 3 | P6 |
| 1 Dominating 15 | P2, P8 | 9 Dominating 4 | P2, P6 |
| 2 Dominating 3 | P8 | 9 Dominating 5 | P2 |
| 2 Dominating 10 | P2, P8 | 9 Dominating 6 | P2 |
| 2 Dominating 11 | P8 | 9 Dominating 10 | P2 |
| 2 Dominating 14 | P4 | 9 Dominating 14 | P7 |
| 2 Dominating 15 | P2, P5 | 10 Dominating 1 | P2, P8 |
| 3 Dominating 7 | P6 | 10 Dominating 3 | P8 |
| 3 Dominating 11 | P1 | 10 Dominating 6 | P2, P8 |
| 3 Dominating 12 | P8 | 10 Dominating 7 | P2 |
| 3 Dominating 13 | P4 | 10 Dominating 8 | P2 |
| 3 Dominating 15 | P3 | 10 Dominating 13 | P2 |
| 4 Dominating 1 | P1, P2, P8 | 10 Dominating 14 | P8 |
| 4 Dominating 2 | P2 | 11 Dominating 6 | P1, P8 |
| 4 Dominating 3 | P1 | 11 Dominating 7 | P7 |
| 4 Dominating 5 | P2, P8 | 11 Dominating 9 | P7 |
| 4 Dominating 6 | P1, P2, P8 | 11 Dominating 10 | P8 |
| 4 Dominating 7 | P2 | 11 Dominating 12 | P2 |
| 4 Dominating 10 | P2, P8 | 11 Dominating 13 | P2 |
| 4 Dominating 11 | P1, P8 | 11 Dominating14 | P8 |
| 4 Dominating 12 | P2, P4 | 12 Dominating 1 | P8 |
| 4 Dominating 13 | P2 | 12 Dominating 2 | P2 |
| 4 Dominating 15 | P2, P4, P8 | 12 Dominating 5 | P2 |
| 5 Dominating 1 | P2, P8 | 12 Dominating 6 | P2 |
| 5 Dominating 2 | P2 | 12 Dominating 7 | P2 |
| 5 Dominating 3 | P8 | 12 Dominating 9 | P2 |
| 5 Dominating 6 | P2, P3, P8 | 12 Dominating 10 | P2 |
| 5 Dominating 7 | P2 | 12 Dominating 13 | P2 |
| 5 Dominating 10 | P2, P8 | 12 Dominating 14 | P4 |
| 5 Dominating 11 | P8 | 13 Dominating 1 | P2 |
| 5 Dominating 15 | P2, P8 | 13 Dominating 2 | P2 |
| 6 Dominating 2 | P2, P8 | 13 Dominating 5 | P2, P3 |
| 6 Dominating 3 | P1, P2, P8 | 13 Dominating 6 | P2, P4 |
| 6 Dominating 7 | P2 | 13 Dominating 8 | P2 |
| 7 Dominating 1 | P2, P7 | 13 Dominating 9 | P2 |
| 7 Dominating 2 | P2, P5, P6 | 13 Dominating 14 | P4 |
| 7 Dominating 8 | P2 | 14 Dominating 3 | P8 |
| 7 Dominating 9 | P2, P6, P7 | 14 Dominating 4 | P8 |
| 7 Dominating 13 | P2 | 14 Dominating 5 | P8 |
| 7 Dominating 14 | P7 | 14 Dominating 6 | P8 |
| 7 Dominating 15 | P2, P5 | 14 Dominating 8 | P5 |
| 8 Dominating 1 | P2 | 15 Dominating 6 | P2, P8 |
| 8 Dominating 2 | P2 | 15 Dominating 8 | P2, P5 |
| 8 Dominating 3 | P6 | 15 Dominating 9 | P2 |
| 8 Dominating 4 | P2 | 15 Dominating 10 | P2, P8 |
| 8 Dominating 5 | P2 | 15 Dominating 11 | P8 |
| 8 Dominating 6 | P2 | 15 Dominating 12 | P2, P4 |
| 8 Dominating 9 | P2 | 15 Dominating 13 | P2 |
| | | 15 Dominating 14 | P4, P8 |

**Table 13.** Dominating interaction matrix.

| CSF | 1 | 2 | 3 | 4 | 5 | 6 | 7 | 8 | 9 | 10 | 11 | 12 | 13 | 14 | 15 |
|-----|---|---|---|---|---|---|---|---|---|----|----|----|----|----|----|
| 1 | | P1,P7,P8 | P2,P8 | P1,2,P8 | P1 | P7 | P1,P8 | P1,P7,P8 | P1,P2,P8 | P1,P7,P8 | P1,P2,P7 | P1,P7,P8 | P1,P7,P8 | P1,P2 | P1,P7 |
| 2 | P2,P6,P5 | | P2,P5,P6 | P2,P5,P6 | P2,P5,P6 | P2,P5,P6 | P6 | P2,P5,P6 | P2,P5,P6 | P2,P5,P6 | P2,P5,P6 | P2,P5,P6 | P2,P5,P6 | P2,P5,P6 | P2,P5,P6 |
| 3 | P1,P5,P6 | P1,P8 | | P5,P6 | P1,P5,P6 | P1,P5,P6 | P1,P5,P6 | P1,P6 | P1,P5,P6 | P1,P5,P6 | P1,P5,P6 | P1,P5,P6 | P1,P5,P6 | P1,P5,P6 | P1,P5,P6 |
| 4 | P8 | P1,P4,P8 | P1,P2,P4,P8 | | P3 | P1,P2,P4,P8 | P1,P2,P4,P8 | P1,P2,P4,P8 | P1,P4,P8 | P1,P4 | P2,P4 | P1,P8 | P1,P4,P8 | P1,P2 | P1 |
| 5 | P2,P3,P8 | P3,P8 | P2,P3,P8 | P2,P8 | | P1 | P8 | P3,P8 | P3,P8 | P3 | P2,P3 | P3,P8 | P8 | P2,P3 | P2,P3 |
| 6 | P1,P2,P3,P8 | P1,P3,P8 | P2,P3,P8 | P3 | P2,P3,P8 | | P1,P2,P3,P8 | P1,P3,P8 | P1,P3,P7,P8 | P1,P3 | P2,P3 | P1,P3,P8 | P1,P8 | P1,P2,P3 | P1,P3,P5 |
| 7 | P5,P6,P7 | P7 | P2,P7 | P5,P6,P7 | P2,P5,P6,P7 | P5,P6,P7 | | P6,P7 | P5 | P2,P5,P6,P7 | P2,P5,P6,P7 | P5,P6,P7 | P5,P6,P7 | P5,P6,P7 | P5,P6,P7 |
| 8 | P2,P5 | P2,P5 | P5 | P5 | P2,P5 | P2,P5 | P2,P5 | | P6,P7 | P5 | P2,P5 | P5 | P3 | P2,P5 | P2,P5 |
| 9 | P2,P6,P7 | P7 | P2,P57 | P2,P6,P7 | P2,P6,P7 | P2,P6,P7 | P2,P6,P7 | P2,P6,P7 | | P6,P7,P8 | P2,P6,P7 | P6,P7 | P6,P7 | P2,P6 | P6,P7 |
| 10 | P2,P8 | P8 | P2,P8 | P2,P8 | P2,P8 | P2,P8 | P2,P8 | P1,P2,P8 | P2 | | P2 | P8 | P8 | P2 | P4,P5 |
| 11 | P1,P8 | P1,P8 | P8 | P1,P8 | P1,P8 | P1,P8 | P1,P8 | P2,P8 | P1,P8 | P1,P8 | | P1,P8 | P1,P8 | P1 | P1 |
| 12 | P2,P4 | P4 | P2,P4 | P2,P4 | P2,P4 | P2,P4 | P2,P4 | P2,P4 | P2,P4 | P2,P4 | P2,P4 | | P2,P4 | P2 | P2 |
| 13 | P2,P3 | P3 | P2,P3 | P2,P3 | P2,P3 | P2,P3 | P2,P3 | P2,P3 | P2,P3 | P2,P3 | P2,P3 | P2,P3 | | P2,P3 | P3 |
| 14 | P4,P7,P8 | P4,P7,P8 | P4,P7,P8 | P4,P7,P8 | P4,P7,P8 | P4,P7,P8 | P4,P7,P8 | P4,P7,P8 | P4,P7,P8 | P4,P7,P8 | P4,P7,P8 | P4,P7,P8 | P4,P7,P8 | | P7 |
| 15 | P2,P4,P5,P8 | P4,P8 | P2,P4,P8 | P2,P4,P5,P8 | P2,P4,P5,P8 | P2,P4,P5,P8 | P2,P4,P5,P8 | P2,P4,P5,P8 | P2,P4,P5,P8 | P2,P4,P5,P8 | P2,P4,P5,P8 | P2,P4,P5,P8 | P2,P4,P5,P8 | P2,P4,P5,P8 | |

**Table 14.** Dominance matrix.

| CSF | 1 | 2 | 3 | 4 | 5 | 6 | 7 | 8 | 9 | 10 | 11 | 12 | 13 | 14 | 15 | (D) * | (D–B) ** | Rank |
|---|---|---|---|---|---|---|---|---|---|---|---|---|---|---|---|---|---|---|
| 1 |  | 3 | 2 | 3 | 1 | 1 | 2 | 3 | 3 | 3 | 3 | 3 | 3 | 2 | 2 | 34 | −4 | 8 |
| 2 | 3 |  | 3 | 3 | 3 | 3 | 3 | 1 | 3 | 3 | 3 | 3 | 3 | 3 | 3 | 40 | 13 | 2 |
| 3 | 3 | 2 |  | 2 | 3 | 3 | 3 | 2 | 3 | 3 | 3 | 3 | 3 | 3 | 3 | 39 | 6 | 4 |
| 4 | 1 | 3 | 4 |  | 1 | 4 | 4 | 4 | 3 | 2 | 2 | 2 | 3 | 2 | 1 | 36 | 3 | 6 |
| 5 | 3 | 2 | 3 | 2 |  | 1 | 1 | 2 | 2 | 1 | 2 | 2 | 1 | 2 | 2 | 26 | −11 | 12 |
| 6 | 4 | 3 | 3 | 1 | 3 |  | 4 | 3 | 4 | 2 | 2 | 3 | 2 | 3 | 3 | 40 | 5 | 5 |
| 7 | 4 | 1 | 2 | 3 | 4 | 3 |  | 2 | 1 | 4 | 4 | 3 | 3 | 3 | 3 | 40 | 3 | 6 |
| 8 | 2 | 2 | 1 | 1 | 2 | 2 | 2 |  | 2 | 1 | 2 | 1 | 1 | 2 | 2 | 23 | −13 | 14 |
| 9 | 3 | 1 | 2 | 3 | 3 | 3 | 3 | 3 |  | 3 | 3 | 2 | 2 | 2 | 2 | 35 | 0 | 7 |
| 10 | 2 | 1 | 2 | 2 | 2 | 2 | 2 | 3 | 1 |  | 1 | 1 | 1 | 1 | 2 | 23 | −12 | 13 |
| 11 | 2 | 2 | 1 | 2 | 3 | 2 | 2 | 2 | 2 | 2 |  | 2 | 2 | 1 | 1 | 26 | −10 | 11 |
| 12 | 2 | 1 | 2 | 2 | 2 | 2 | 2 | 2 | 2 | 2 | 2 |  | 2 | 1 | 1 | 25 | −9 | 10 |
| 13 | 2 | 1 | 2 | 2 | 3 | 2 | 2 | 2 | 2 | 2 | 2 | 2 |  | 2 | 1 | 27 | −6 | 9 |
| 14 | 3 | 3 | 3 | 3 | 3 | 3 | 3 | 3 | 3 | 3 | 3 | 3 | 3 |  | 1 | 40 | 9 | 3 |
| 15 | 4 | 2 | 3 | 4 | 4 | 4 | 4 | 4 | 4 | 4 | 4 | 4 | 4 | 4 |  | 53 | 26 | 1 |
| (B) *** | 38 | 27 | 33 | 33 | 37 | 35 | 37 | 36 | 35 | 35 | 36 | 34 | 33 | 31 | 27 |  |  |  |

* Number of cases dominating, ** Net dominance, *** Number of cases being dominated.

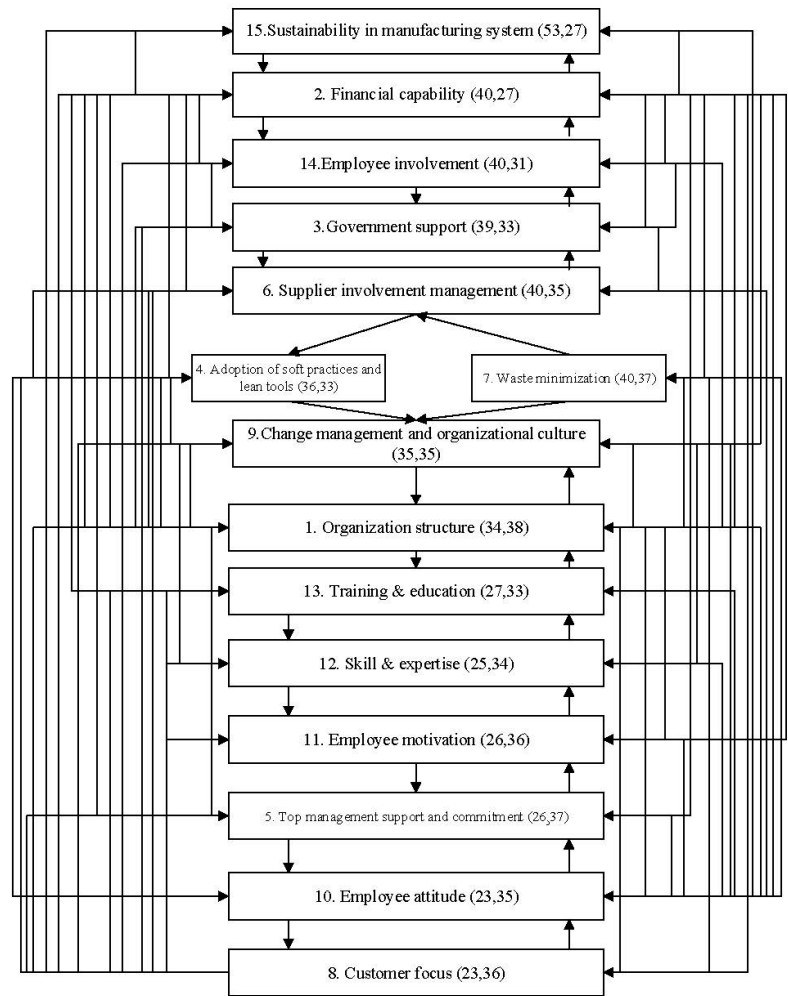

**Figure 5.** Interpretive ranking process model.

## 5. Discussion

In this fiercely competitive market, SMEs are trying to maintain their market share by implementing various initiatives such as updating equipment, implementing new IT systems, and implementing novel manufacturing systems. Lean manufacturing implementation can help these SMEs reduce their manufacturing costs and increase the value of the existing products, thus gaining a cutting edge on their rivals in terms of business-level performance and competitiveness [60,61]. The first objective of the present research has been accomplished using an in-depth review of the literature. Twenty CSFs have been shortlisted with the help of an expert group, which was reduced to fifteen using the questionnaire feedback. Practicing managers may concentrate on the most relevant criteria that influence lean implementation. Practicing managers must take care while assessing relevant criteria. Any potential criteria missed in the beginning stage may delay the lean implementation process. The CSF of "top management" is regarded as a driving force in any organization because customers' needs are met with the right vision and mission [18].

The second objective of the present research was accomplished using a combinatorial approach of ISM and MICMAC analysis. The lean implementation CSFs need to be evaluated based on their prevailing contextual relationships. Thus, the role of each CSF can be investigated at the prior stage before actually implementing lean in the manufacturing sector. ISM plays a vital role in identifying the relationship, which is necessary for relationship modeling. Using the ISM, practicing managers will be able to visualize the influence of each CSF on another. The CSF may drive another CSF or may be driven by another driving CSF, so controlling such depending CSF will be manageable. The knowledge of influence will be useful in controlling the CSF. Thus, allocating resources and optimizing them will be easy, which is the main concern of practicing managers and management. The digraph revealed that "government", "financial capability", and "top management support and commitment" play a significant role in lean implementation for SMEs. The findings are in line with previous research claiming that management leadership and its financial capability are crucial [3]. The study by [16] claims that "workforce skill and expertise" play a vital role in lean implementation. The study also supports the obtained results, which suggest that "training and education", and "resources and capability" are useful. As per the study conducted by Houti et al. [14], "top management support and commitment" and "adoption of lean soft practices and lean tools" play a contributing role in attaining much-needed manufacturing sustainability. They help in the manufacturing system through "employee involvement", "supplier involvement management", utilizing skills and expertise to have more "customer focus", and "waste reduction". Further, these CSFs provide long-term development and manufacturing sustainability [54].

Further, this process will help in cutting down the gestation period while implementing a lean manufacturing system. MICMAC analysis helps in grouping the CSF into four clusters. The CSFs may be grouped into four clusters: dependent, independent, autonomous, and linkage categories. The use of MICMAC analysis offers a way to investigate each lean CSF. Looking at the MICMAC analysis of the lean CSF for the manufacturing sector, it is seen that there are no lean implementation CSFs or resulting CSFs in the autonomous and linkage cluster. CSF4 "adoption of soft practices and lean tools", CSF6 "supplier involvement management", CSF7 "waste minimization", CSF8 "customer focus", CSF12 "skill and expertise", CSF14 "employee involvement", and CSF15 "sustainability in manufacturing system" exhibit high dependence and weak driving power; hence they are classified as dependent cluster,, whereas CSF1 "organization structure", CSF2 "financial capability", CSF3 "government support", CSF5 "top management support and commitment", CSF9 "change management and organizational culture", CSF10 "employee attitude", CSF11 "employee motivation", and CSF13 "training and education" exhibit high driving power and weak dependence, thus classified as an independent cluster. Hence, it can be concluded that all of the factors of lean implementation are stable. There is no CSF in autonomous or linkage clusters. This will further help practicing managers by providing proactive actions before a lean implementation is carried out. Looking at the independent CSFs, various

activities, such as training the employees, skill enhancement, knowledge creation, and resource allocation, may be planned.

The third objective has been fulfilled using IRP. The categorization of CSFs into clusters helps in controlling CSFs. The ranking also adds further value in taking proactive action to control CSFs. Further, this knowledge will be helpful as lean manufacturing may deliver different results depending on the level of CSFs implementation [8]. The ranking of CSFs may provide essential information for successful lean implementations. The ranking is also beneficial to lean manufacturing stakeholders. Employers and employees will be aware of the scope of each CSF. Because there is a risk of bias in decision making, the IRP could be utilized to replace the AHP process. The use of IRP reduces the chances of such decision biases; hence the decision-making accuracy is improved. The interpretative ranking revealed that "sustainability in a manufacturing system", "financial capacity", "employee involvement", "government support", and "supplier involvement management" play a vital role in successful lean implementations. Upon carefully observing these CSFs, it may be concluded that apart from the financial capability of SMEs, the sustainability of the manufacturing sector is vital, which can be attained by implementing lean in the manufacturing sector of SMEs. The findings of the present study match with the previous studies [29,57]. Practicing managers will be able to know the rank of each CSF, which will help in identifying the right strategy for the firm. The strategic action by the stakeholder group could become timely and well-controlled concerning various parameters such as time, cost-competitive, challenges, delays, etc.

The present study has geographic limitations. The CSFs involve "employee attitude", "employee motivation", "skill and expertise", and "training and education" may vary from region to region. The same CSFs in another part of the country will pose different challenges, which will make the agendas for lean implementation costly and time-consuming. The lean implementation CSFs also influenced by government laws and policy. Though the lean implementation CSFs may differ from sector to sector of SMEs, a partial generalization may be drawn from such relationship modeling for future implementation. The respondents' approach to CSFs may also vary under different situations.

## 6. Conclusions

The present research examines the lean implementation CSFs using the ISM, MICMAC, and IRP. ISM helps in quantifying the influence of each CSF with other CSFs using contextual relationships, whereas MICMAC helps in classifying into the cluster. The outcomes of both methodologies are useful to practicing managers for understanding the CSF before the actual lean implementation. Further, the relation modeling reveals a significant relationship among the lean implementation CSFs, which helps devise strategy and decision making. The findings of this study are relevant in terms of lean implementation in manufacturing SMEs. Understanding each CSF will aid the industry in controlling and completing them throughout lean deployments.

The IRP methodology employed in this research has two major strengths: (a) the extent of dominance information is not mandatory, and (b) the methodology offers ease in measuring and comparing the influence of interaction. Ranking of the CSFs provides liberty to practicing managers to prioritize strategies and decision making. The SME's decision-making process will become more flexible and fast as compared to large enterprises, and the early control of such CSFs will help to maneuver the success of lean implementations. The various sectors may adopt the ISM, MICMAC, and IRP approaches to reveal the crucial relationship among lean CSFs for successful lean implementation. The natural extension of this work may be extended into different sectors to reveal common CSFs. The exploratory analysis along with structural equation modeling may provide some interesting results.

**Author Contributions:** Conceptualization, K.M.Q., M.R.N.Q. and B.G.M.; methodology, K.M.Q., M.R.N.Q., M.M.; software, M.M., M.R.N.Q.; validation, K.M.Q., M.R.N.Q. and B.G.M.; formal analysis, M.M., S.Y.A. and N.A.; investigation: B.G.M., S.Y.A. and N.A. Resources: S.Y.A. and N.A.; data curation, K.M.Q.; writing—original draft preparation, K.M.Q., B.G.M.; writing—review

and editing, M.R.N.Q.; visualization, S.Y.A. and N.A.; supervision; S.Y.A., N.A., B.G.M.; project administration: S.Y.A., N.A. and M.M.; funding acquisition, S.Y.A. and N.A. All authors have read and agreed to the published version of the manuscript.

**Funding:** This research was funded by the Deanship of Scientific Research, King Khalid University, Kingdom of Saudi Arabia, and the grant number is R.G.P.1/212/41.

**Institutional Review Board Statement:** Not applicable.

**Informed Consent Statement:** Not applicable.

**Data Availability Statement:** Not applicable.

**Acknowledgments:** We would like to express our gratitude to the Deanship of Scientific Research, King Khalid University, Kingdom of Saudi Arabia, for funding this work, as well as family, friends, and colleagues for their constant inspiration and encouragement.

**Conflicts of Interest:** The authors declare no conflict of interest.

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
