# Peer review of "Accomplishing Sustainability in Manufacturing System for Small and Medium-Sized Enterprises (SMEs) through Lean Implementation"

_sustainability, doi:10.3390/su14159732_

Round 1
Reviewer 1 Report
This manuscript presents an evaluation of the critical success factors (CSFs) of lean implementation (LI-CSFs) using the Interpretive Structure Modeling (ISM) approach. In general, the article looks raw, and the research design is entirely unclear. I suggest the authors rewrite this article (literally rewrite, not merely revise) to deliver a more readable presentation of their research processes and results. The following comments intend to help the authors during the rewrite of this manuscript.
ABSTRACT.
- Line 21. Before explaining the MICMAC analysis, please add a couple of sentences on how the authors identify the CSFs of Lean Implementation.
- Line 27. After explaining the core findings, please add a couple of sentences to highlight the policy/managerial implications of this study based on the findings.
INTRODUCTION.
- Since this is a scientific paper, please elaborate the introductory narration by referring to validated/peer-reviewed literature. Instead of merely putting references at the end of existing sentences, the authors should explain what each literature talked about in the discussed point/issue. The cited literature should be relevant and support all points/issues in this "Introduction" section.
METHODOLOGY.
- This manuscript lacks a "Methodology" section to present the step-by-step of the research (from the "Identification of Lean Implementation CSFs" to the "Interpretative Ranking Process"). Before "Literature review to identify the Lean Implementations CSFs", the authors should add a new Section 2 (Methodology) to explain the design of the entire research. This section would convince readers that this study was systematically conducted, and has included all necessary activities.
- The explanations of the research design are currently scattered throughout the articles. Consider adding the following lines into this new section: lines 88-108 (do not forget to explain Figure 1), 131-132 ('The reliability and ... 26.0 software)."), 134-136 ("To keep the reliability ... sent for feedback."), 138-141 ("The corrected item- ... test was repeated."), 162-177, 233-239 ("The reachability element ... lowest level."), 261-265 ("The structure ... each CSFs."), 329-331, and 390-408.
- LI-CSFs search. The authors should elaborate further on how they searched and found references in Table 1. The measures of the search must be clear, and the search protocol must be explained in detail. Otherwise, the references appear from an arbitrary process, reducing the validity of literature selection and the adequacy of the chosen literature.
RESULTS.
- Since the current sections 2-5 present the results of this study in various research stages (as per the "Methodology" section suggested above), these sections (currently 2-5) should be treated as a single Section 3 (Results). The existing sections should become the subsections of the new "Results" section. For any explanations regarding the research steps/stages, they should be moved to the suggested "Methodology" section for clarity purposes.
- Table 2. The "descriptions" presented have made the table unnecessarily long. Instead of putting all "descriptions" in the table, the authors should make use of the descriptions for elaboration in the text. It would present the thought process of the authors leading to the discovery of the Lean Implementation CSFs (Table 2, column 2), enriching the entire "Literature Review" section.
- Line120-125. The authors should explain who the respondents were, where did they come from, and how did the authors select them. Without any explanations on these three issues, the respondents appear from nowhere.
- Line 130. Please rename this subsection to avoid a misleading reference to the core analysis presented later.
DISCUSSION (currently "Results and Discussion").
- Line 444. Since the real results of the research stages have been presented before, please rename this into the "Discussion" section.
- Because this study focused on three objectives (lines 67-70), this "Discussion" section should be divided into three subsections, in which each explains findings relevant to one specific research objective. It would convince readers that this study has precisely accomplished its objectives.
- Besides, please elaborate more on the discussion in lines 445-449 and lines 463-476 by referring to relevant literature. Instead of merely putting references at the end of existing sentences, the authors should explain what each literature talked about in the discussed finding.
CONCLUSION.
- Please put lines 485-490 to the end of the "Conclusion" section (without the title/line 484).
- Please move any citing sentence in this section (lines 495-497) to the "Discussion" section.
- Then, please enrich this section by dividing it into three core parts. Consider making one paragraph for each part.
- Part 1: Summary of this study.
- Part 2: Present key findings that fulfill each research objective briefly. This is proof that this study successfully accomplishes its objectives.
- Part 3: Suggest managerial/policy implications based on the key findings of this study.
Author Response
To
Reviewer/Editor
Thank you very much for your valuable time and comments, which have enhanced the quality of the present manuscript considerably.
Authors
Date: 23rd June 2022

Reviewer 2 Report
Accomplishing Sustainability in Manufacturing System for 2 Small and Medium-sized Enterprises (SMEs) through Lean Implementation
Review
The authors present a interesting MICMAC approach to evaluate lean manufacturing. While I cannot comment on the theoretical justification of the scales, I want to applaud the authors for their clear writing and the well-thought-out approach. Therefore, I can only provide some comments that the authors should evaluate. Most of them have to do with the presentation of the figures and the tables. I'll provide a few examples below, but these are not the only ones.
- What do the arrows represent in Figure 5?
- Tables and figures are sometimes poorly captioned. Ensure the content is understandable without the main text.
- Table 14 should be revisited. Is this necessary for the main story or could it be in an appendix. In any case the current formatting should be rethought.
- Section 7 should be expanded. While the work is very well done, the authors could be a bit more critical of themselves and of things to do in the future that did not fit in the current work.
- Also, there are quite a few keywords which makes the focus of the paper a bit difficult to define.
Author Response

(The authors gave the same response as above.)

Reviewer 3 Report
The article concerns an interesting issue of implementing selected lean tools in small and medium-sized enterprises to increase the efficiency of implemented processes, especially in production. The study is relevant both for the needs of small and medium-sized enterprises and as an inspiration to undertake scientific research in this area. However, I have a number of doubts and comments that should be made.
1) First of all, I have doubts about the title of the article, which suggests that by implementing lean tools in small and medium-sized enterprises it will allow them to achieve sustainable development - I absolutely cannot agree with this, companies may achieve better economic results, production results, etc., but they will not necessarily achieve sustainable development, we can only mark "in the way to sustainable development…".
2) In the introductory part, the problem should be more articulated, why it is important and what solutions can be proposed in relation to it, why the authors have undertaken such a research problem.
3) No clearly outlined research methodology, no research questions and hypotheses, only poorly outlined objectives of the article.
4) Chapter 2 should include the limitations of existing work and how this study contributed to addressing those limitations.
5) Each chapter should include a brief introduction to its content.
6) Figure 1 needs to be corrected; it is illegible and detracts from the presentation of the content throughout the paper.
7) There are too few references in the discussion to current work of other authors, especially in the second part of the chapter.
8) Very weak conclusions, need to be supplemented.
9) More authoritative literature from 2021 and 2022 should be added to the references.
Author Response

(The authors gave the same response as above.)

Reviewer 4 Report
The article is correct in terms of content. The article deals with current issues related to Lean Manufacturing, Sustainable Development, LSS and CSF. The developed model is correct. The conclusions are right. The selection of literature is sufficient. These are the positive aspects of the study.
Potentials for improvement:
-) I propose to add the purpose of the article in the abstract.
-) There are no references to literature in the 1st Introduction. In my opinion, it is necessary. I propose to supplement the references to literature.
-) Table 1. Please provide the date from which the literature analysis was performed. Has it been since 2006? If so, why from this year?
-) Please correct figure 1. Half of the word "manufacturing" is invisible. The other drawing blocks (fields) also do not show the full content.
-) In my opinion subchapter 2.1 should start on line 83.
-) I suggest not to end the subsection with a drawing only with the summary text (e.g. line 108)
-) Line 129, 258 and others. After such an extensive table, it is necessary to add conclusions from its presentation, before the end of sub-chapter 2.1. Similarly in the case of chapter 3.
-) Line 149. Mistake in table numbering. This should probably be Table 3.
-) Lines 160 and 161. What are these wide unfilled lines for?
-) Line 186 and 88. Please check and correct the numbering of the tables until the end of the study.
-) Table 7. IRM. , Table 8. FRM. - Please expand the acronyms in the titles.
-) Brak powołania w tekście na tab. 15. Powinno być przed tabelą.
This research was funded by the Deanship of Scientific Research, King Khalid Uni-512 versity, Kingdom of Saudi Arabia, and the grant number is R.G.P.1 / 212/41.
I don't feel qualified to judge about the English language and style.
Overall, I recommend that you publish the article after corrections have been made.
Author Response

(The authors gave the same response as above.)

Round 2
Reviewer 3 Report
The submitted article is very unreadable, I find it hard to relate to the content introduced. A well-developed research methodology is still missing, as well as additions to the literature survey.
Author Response
To
Reviewer/Editor
Thank you very much for your valuable time and comments, which have enhanced the quality of the present manuscript considerably.
Authors
Date: 21st July 2022
